# From single decisions to sequential choice patterns: Extending the dynamics of value-based decision-making

Stefan Scherbaum[1]*, Steven J. Lade[2,3,4], Stefan Siegmund[1], Thomas Goschke[1], Maja Dshemuchadse[5]

1 Technische Universität Dresden, Dresden, Germany, 2 Max Planck Institute for the Physics of Complex Systems, Dresden, Germany, 3 Stockholm University, Stockholm, Sweden, 4 The Australian National University, Canberra, Australia, 5 Hochschule Zittau-Görlitz, Görlitz, Germany

* Stefan.Scherbaum@tu-dresden.de

**Data Availability Statement:** Primary data (csv format), analysis files (Matlab) and model code (Matlab) are available via the Open Science Framework (osf.io/d2yhb).

## Abstract

Every day, we make many value-based decisions where we weigh the value of options with other properties, e.g. their time of delivery. In the laboratory, such value-based decision-making is usually studied on a trial by trial basis and each decision is assumed to represent an isolated choice process. Real-life decisions however are usually embedded in a rich context of previous choices at different time scales. A fundamental question is therefore how the dynamics of value-based decision processes unfold on a time scale across several decisions. Indeed, findings from perceptual decision making suggest that sequential decisions patterns might also be present for vale-based decision making. Here, we use a neural-inspired attractor model as an instance of dynamic models from perceptual decision making, as such models incorporate inherent activation dynamics across decisions. We use the model to predict sequential patterns, namely oscillatory switching, perseveration and dependence of perseveration on the delay between decisions. Furthermore, we predict RT effects for specific sequences of trials. We validate the predictions in two new studies and a reanalysis of existing data from a novel decision game in which participants have to perform delay discounting decisions. Applying the validated reasoning to a well-established choice questionnaire, we illustrate and discuss that taking sequential choice patterns into account may be necessary to accurately analyse and model value-based decision processes, especially when considering differences between individuals.

## Introduction

Value-based decision-making is a central part of our human existence, often asking for a trade-off between different interests: Choosing between hearty or healthy meal options, between current joys and pension plans, or–as folk-wisdom names it–between one bird in the hand or two in the bush. Studies of value-based decision-making have provided profound insight on which information decision-makers base such choices [1–5]. In such studies,

**Funding:** SSCH: German Research Council (DFG) (grant SFB 940/3 2020 Project A8) SL: Swedish Research Council Formas (Project Grant 2014-589). The funders had no role in study design, data collection and analysis, decision to publish, or preparation of the manuscript.

**Competing interests:** The authors have declared that no competing interests exist.

participants usually have to choose between two options which offer a trade-off. For example, in delay discounting studies, participants have to choose between one option which is small, but delivered soon, and another option that is large, but delivered later. Since such decisions are about the weighing of the different attributes of options, e.g., their monetary value, their temporal delay or probability, and these attributes are assumed to be integrated into a common value for each option, such decisions are called value-based decisions.

The behavioural and neural data from value-based decision-making studies is usually acquired in a trial-wise fashion and results are analysed in a way as if these trials would be independent from one another. While such an approach matches common decisions models, e.g. sequential sampling models that conceive of the decision process as a noisy process of information accumulation leading to a final choice [6–10], it neglects the inherent dynamics over sequences single decisions. Hence, potential patterns that extend across multiple decisions [11–16] are usually ignored. However, such patterns have been found to be ubiquitous in perceptual decision making [e.g. 17–20] and hence should also not be ignored in value-based decision making. First evidence suggests, that value-based decision making also shows such patterns [21, 22]. Interestingly, such patterns are typical for non-linear coupled models, such as leaky integrator models [23] or parallel constraint satisfaction models [24, 25], which is why we use one instance of this class of models, a simple neural-inspired attractor model of decision-making, which incorporates the inherent activation dynamics of such models [11, 23, 26] to predict typical choice patterns in sequences of values-based decisions, namely oscillatory switching (i.e., the probability distribution for switching back and forth between choices), perseveration (sticking to a choice more often than switching), and dependency on the temporal distance between decisions (higher perseveration when decisions closely follow each other) and response time (RT) patterns in such sequences. We will present empirical data from three experiments that show the predicted sequential patterns and will finally perform a modelling study that shows that such sequential patterns might bias value-based decision research if ignored.

## Model predictions

We predicted sequential choice patterns across sequences of choices and RT effects for specific trials in these sequences using a neural-inspired attractor model (see Fig 1a) that we used

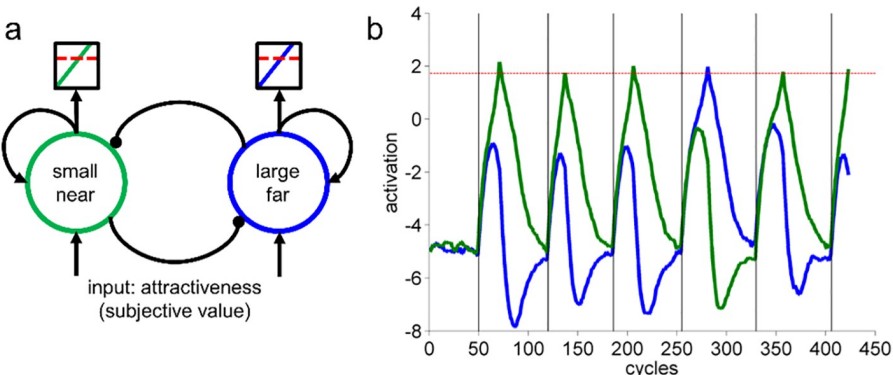

**Fig 1. The neural attractor model.** a) Sketch of the neural-inspired attractor model. Each units' activation represents the attractiveness of one option and inhibits the activation of the alternative option. When an option's activation reaches the choice-threshold, the choice of this option is elicited. b) Activation flow across six consecutive choices. After choosing the (here more attractive) small-near option in the first trial, the system perseverates with this choice and only occasional switches to the alternative can be observed, which are caused by noise in the system.

before to predict path dependence [11], and biases from trial to trial [22]. In the following, we present the model briefly on a conceptual level and present the results of computational simulation of the model, the details of which can be found in the appendix (S1 Appendix) and elsewhere [11].

The model takes the attractiveness of each option as input to two simple neural network units representing these options and competing with each other. A unit representing the option with a higher attractiveness hence receives a stronger input and has a higher chance of reaching an activation threshold first, which elicits the choice (similar to sequential sampling models). Notably, noise in the system might still push the less attractive unit across the threshold, leading to a choice of this option. After eliciting a choice, the input is switched off and the units' activation falls back to resting level following the intrinsic dynamics of the system.

If new options are presented while the recently winning unit still shows residual activation, then this option gains a relative advantage in the next choice process (see Fig 1b). We used numerical simulation to derive predictions of sequential choice patterns from the model (see S1 Appendix for more details). The first pattern predicted by the model is noise-induced oscillatory switching as measured by the distribution of switches between options in a sequence of repeating choices (see Fig 2a). First, if the more attractive option is chosen initially, it is either chosen consistently or–if a switch occurs due to noise–chosen again quickly. This leads to 0 switches or an even number of switches—back and forth—between the attractive and the unattractive option. Second, if the unattractive option is chosen initially–due to noise–subjects quickly return to the attractive option. This leads to one switch or an odd number of switches. For options that are equally attractive, there is random distribution of switches across sequences of choices that is not biased to an even or odd number of switches. This prediction might seem trivial but should be seen as a basic sanity check of the model's predictions in real value-based choice data. The central patterns of interest are the next two ones. The second pattern predicted by the model is perseveration as measured by Markov Analysis (see Fig 2b): A choice of an option increases the probability of repeating this choice independent from the

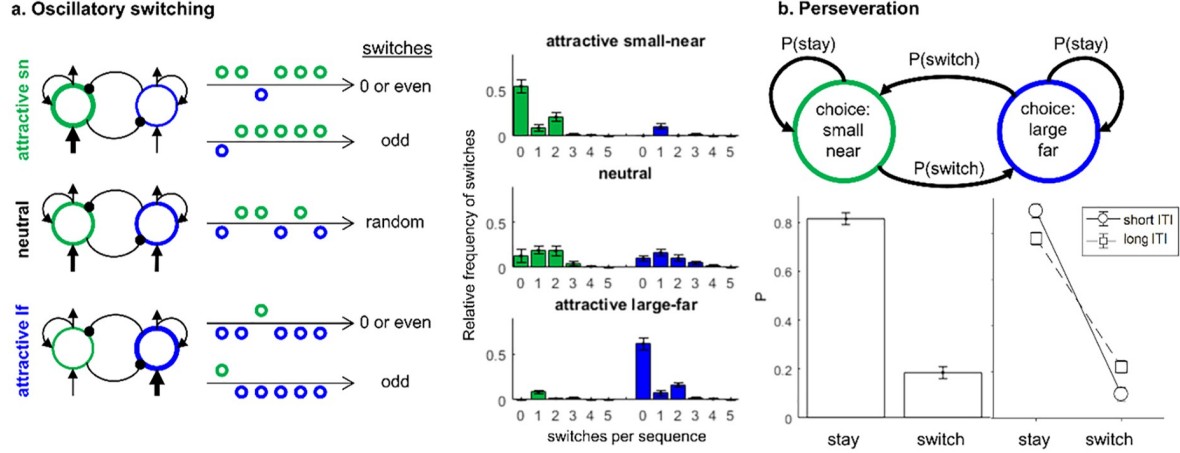

**Fig 2. Model predictions for sequences of repeating decision situations.** a) The model predicts patterns of oscillatory switching that depend on the options' attractiveness and the initial choice in sequences of choices. If the attractive option is chosen initially, the model predicts that zero (staying with the attractive option) or an even number of switches across the sequence are more likely than an odd number of switches. If the unattractive option is chosen initially, the model predicts one (switching back to the attractive option) or an odd number of switches. b) The model predicts choice perseveration as identified by markov analysis: Choosing an option in one trial leads to a higher probability of staying with this choice in the next trial. This perseveration effect is stonger for short inter trial intervals (ITI) than for long ITI. Error bars mark 95% confidence intervals.

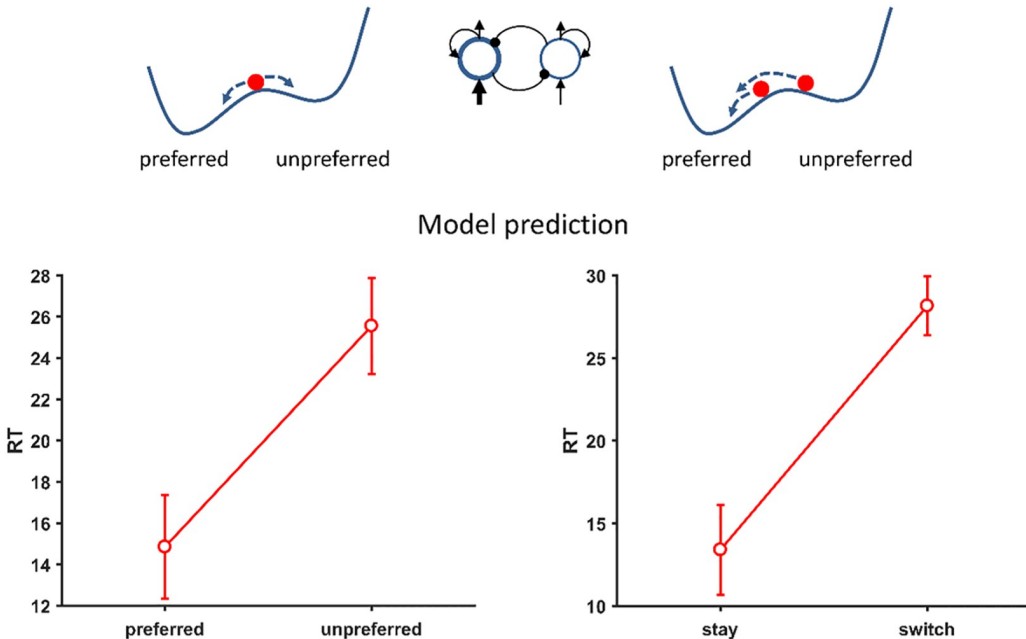

**Fig 3.** Model predictions for RT effects (in cycles) for specific trials in sequences of choices Left: When choosing between two options in trials with clearly given preferences (an attractive SN or attractive LN option), choosing the preferred option should be easier than choosing the unpreferred option. Right: In trials with clearly given preferences and choices of this preferred option, it should be easier to choose the preferred option repeatedly (staying with the preferred option) than to come from the unpreferred option (switching to the preferred option).

relative attractiveness of the options. The third pattern is a temporal dependency of perseveration: Separating two decisions by a short delay compared to a longer delay should lead to stronger perseveration (see Fig 2b).

We further predicted RT effects for specific trials in sequences of choices. First, when choosing between two options in trials with clearly given preferences, choosing the preferred option should be easier than choosing the unpreferred option (Fig 3 left). Second, in trials with clearly given preferences and choices of this preferred option, it should be easier to choose the preferred option repeatedly (staying with the preferred option) than to come from the unpreferred option (switching to the preferred option, Fig 3 right).

## Experiment 1

To study the effects predicted by the model empirically, we used a gamified decision task in which participants had to navigate an avatar in a virtual world to collect coins, choosing between small coins near to their avatar and large coins farther away within limited time [27]. This task was used previously to study sequences of choices [11] and offers the benefit that the spatial nature of the task allows to present the options at changing position, masking the sequential nature of the choices. We expected to find the predicted patterns of oscillatory switching depending on the initial choice (Hypothesis I) and perseveration overall (Hypothesis II) and we expected the predicted RT effects, namely shorter RT for choosing the preferred option (Hypothesis III) and shorter RT for staying with the preferred option (Hypothesis IV; for the modulation of perseveration by the delay between consecutive decisions, please see experiment 3).

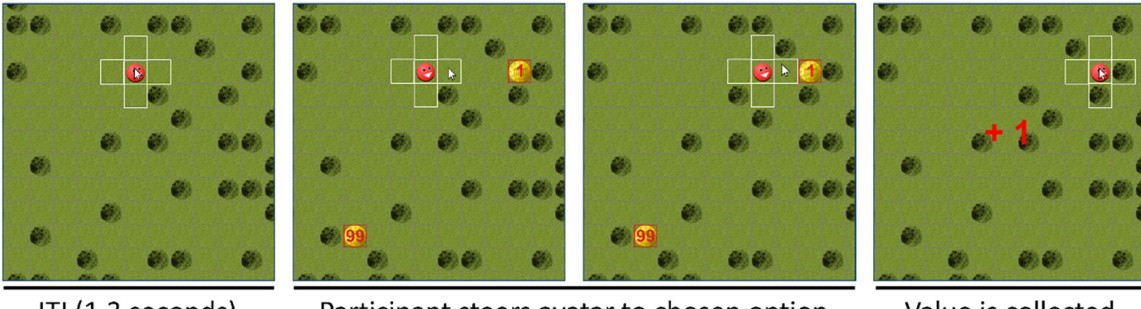

**Fig 4. Sketch of one trial in the delay discounting game.** During the inter-trial-interval of 1.3 seconds, the mouse cursor is locked in the avatar's current position. After the ITI, the two options are presented as coins with values written on top in red font. Participants have to steer the avatar field-by-field to the option they want to collect. When they reach the option's field, the option's value is collected and the collected value is presented to indicate that the choice is completed. At all time, participants could see the remaining time within one block and the collected credits in Euro on a panel at the right side of the playing field.

## Methods

**Ethics statement.** The study was performed in accordance with the guidelines of the Declaration of Helsinki and of the German Psychological Society. An ethical approval was not required since the study did not involve any risk or discomfort for the participants. All participants were informed about the purpose and the procedure of the study and gave written informed consent prior to the experiment. All data were analysed anonymously.

**Participants.** 15 participants of the Technische Universität Dresden took part in the experiment. All participants had normal or corrected-to-normal vision. They gave informed consent to the study and received either payment or class credit.

**Setup and procedure.** Participants' moved an avatar on a playing field divided into 20 x 20 fields (see Figs 4 and 5a). To move the avatar, participants clicked with the mouse in one of four horizontally or vertically adjacent movement fields, as signalled by a white border

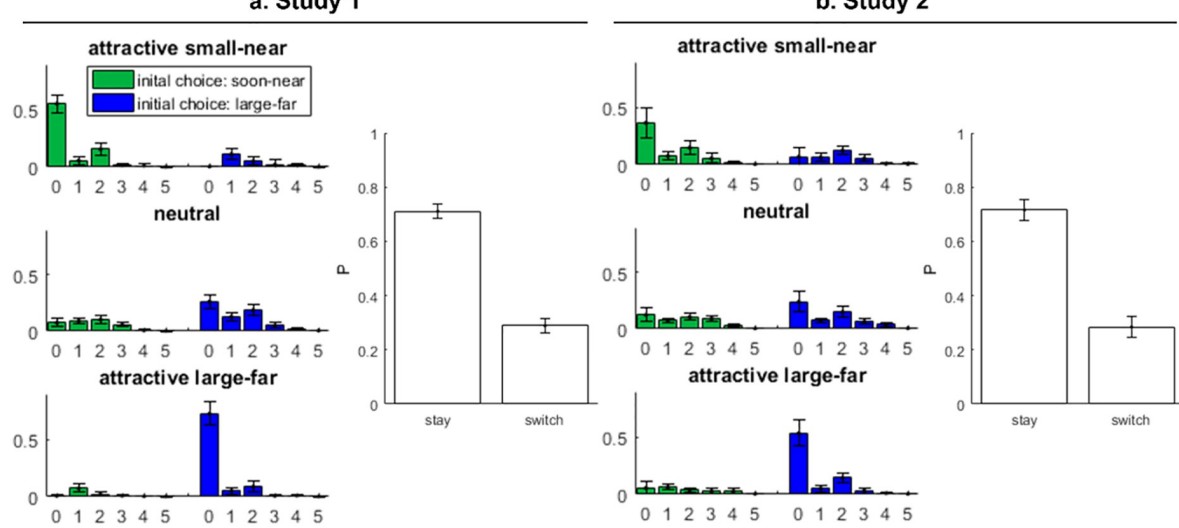

**Fig 5. Choice results of studies 1 and 2.** a) Results of study 1 show oscillatory switching in relative switch frequencies (left sub-panel) and choice perseveration (right sub-panel) as predicted by the model. B) Results of study 2 confirm the results of study 1. Error bars mark 95% confidence intervals.

surrounding the fields. Their task was to collect as much credits as possible within the allotted time limit. On each trial, two options were presented as coins, with each option's credit *value* printed as a red number on the coin. The horizontal and vertical distance of each option (field by field) to the avatar represented the *distance* of the option. One option was near but small (SN), the other option was far but large (LF). Options' *values* ranged from one to ten credits and *distance* ranged from two to fifteen fields. To collect an option, participants navigated the avatar to the chosen coin. When the avatar reached the coin, both options disappeared and the value of the selected option was shown again to the participant as a feedback that the trial was complete and the option's value was collected. After each trial, participants had to wait for the next trial in an inter trial interval (ITI) of 1.3 seconds. Within this interval, the mouse cursor was locked in the centre of the field containing the avatar. After the ITI, the next trial started. At all times, participants could see the remaining time within one block and the collected credits in Euro (1 credit = 1/10 € cent) on a panel at the right side the playing field.

The experiment consisted of three blocks, with one block lasting eight minutes. This amount of time allowed participants to work through a complete design matrix of trials (see below) at least one time. Between blocks, participants were informed about the credits collected and were instructed to rest briefly before the self-paced start of the next block.

Before the start of the experimental blocks, participants worked through a test block of one minute to get used to the virtual environment and handling of the mouse.

**Design.** In the game, participants had to choose between consecutively presented similar choices: distances of the SN option ($D_N$ = [2, 5]) and the LF option ($D_F = D_N$ + [1, 4, 7]) were orthogonally varied in random order across trials; credit values of the SN option ($V_S$ = [10, 20, 30, 40, 50]) and the LF option ($V_L$ = [100, 90, 80, 70, 60]) were combined pairwise (10 and 100, 20 and 90, and so on) and kept constant within each sequence. Sequence length varied randomly between 4, 5, and 6 choices. This yielded 30 sequences of an average length of 5 trials; 150 trials in total for a complete block. We created more blocks than participants could perform within the limited time frame (3 x 8 minutes) to be sure that participants had enough trials to work through.

## Results

Hypothesis I stated that in sequences of trials that start with a choice of the attractive option, we expect no switches or an even number of switches. In contrast, in sequences of trials that start with a choice of the unattractive option, we expect an odd number of switches. For trials with no clearly attractive option, we expected a random distribution of switches unbiased to an odd or even number of switches.

To identify sequences with attractive SN or LF options or with no clearly attractive option, we calculated for each participant the individual indifference point [e.g. 27] and then sorted sequences according to distance of the ratio of the SN and LF option's value to the indifference point. For example, if a participant had an indifference point of 0.5, this means that an SN option with a value ½ of the LF option (e.g. SN: 5 credits, LF: 10 credits) is equally attractive to this participant because of discounting. For this participant, an SN option's value with only 0.3 of the LF option's value (e.g. SN: 3 credits, LF: 10 credits), would lead to a distance to the indifference point of -0.2. For each participant, we calculated the median of distances to the indifference point and categorized all sequences according to their distance to the indifference point as attractive SN, neutral, and attractive LF. For these categories, we counted the number of sequences that started either with the choice of the SN option or the LF option and that yielded either an even or an odd number of switches between options. For statistical analysis, we subtracted the number of odd switches from the number of even switches. The resulting

difference score was analysed in a repeated measures analysis of variance (RMANOVA). This revealed a significant main effect of the initially chosen option (SN vs. LF), $F(1,14) = 7.827$, $p = .014$, $\eta^2_p = .359$, a significant main effect of attractiveness (SN vs. neutral vs. LF), $F(2,28) = 11.420$ $p < .001$, $\eta^2_p = .449$, and–as predicted–a significant interaction, $F(2,28) = 270.036$, $p < .001$, $\eta^2_p = .951$. Hence, we found the predicted pattern of a largely higher number of even switches for sequences in which the attractive option was chosen initially.

Hypothesis II stated that we should find a perseveration bias, which is that the probability to choose the option again that has been chosen previously, is higher than the probability to switch between options. We performed a markov analysis across all trials, in which we counted the number of SN and LF choices with respect to the previously chosen option. This markov analysis takes into account the baseline number of choices of the SN/LF option so that perseveration cannot simply occur because of a general biased to the SN/LF option. For statistical analysis, we analysed whether the difference of switch stay probabilities was positively different from zero, using a dependent samples $t$-test, which revealed a significant difference from zero, $t(14) = 16.938$, $p < .001$, $d = 4.373$. Hence, a clear perseverative bias was present as predicted by the model. A graphical summary of the results is given in Fig 5a.

Hypothesis III stated that choosing the preferred option should be faster than choosing the unpreferred option. We defined RT as the time from the presentation of the options until the first click into a movement field, a measure we previously called decision time [11, 27]. We performed a t-test on the dependent variable RT with the independent variable *choice of preferred-unpreferred option* in trials with one attractive option, which yielded a significant difference, $t(14) = 3.62$, $p = .003$, 95% CI [.033 sec, .131 sec] (Fig 6 top-left).

Hypothesis IV stated that choosing in trials, where participants stay with the preferred option, they are faster than in trials where participants switch to the preferred option (when

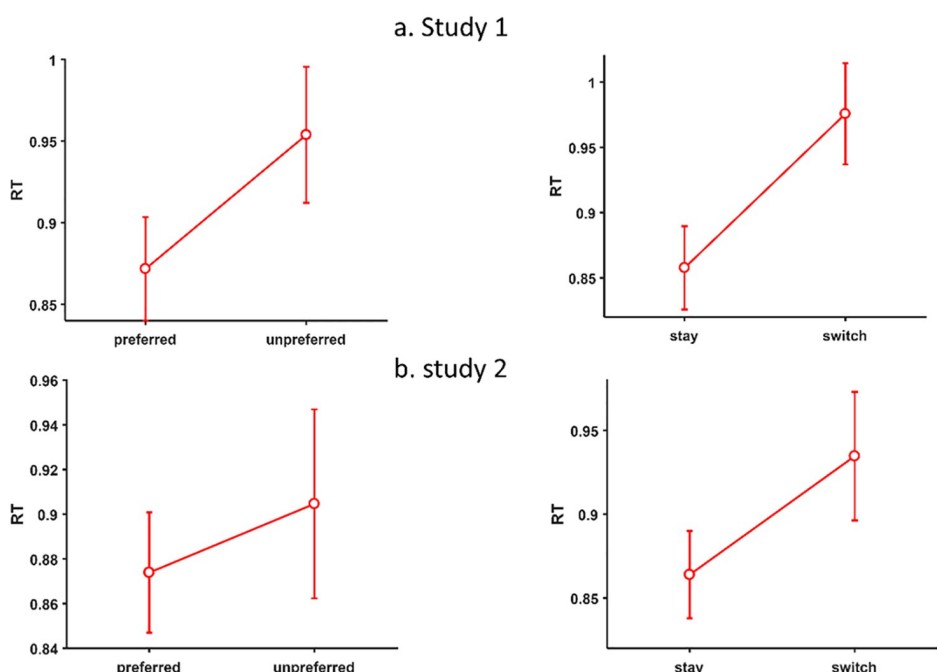

**Fig 6. Results of studies 1 and 2 for RT (in seconds) for trials in which participants choose the preferred vs. the unpreferred option (left) and for trials in which participants stayed with the preferred option vs. switch from the unpreferred to the preferred option (right).**

they chose the unpreferred option in the previous trial). We performed an t-test on dependent variable RT with the independent variable *stay with/switch to preferred option* in trials with one clearly attractive option, which yielded a significant difference, $t(14) = 4.76$, $p < .001$, 95% CI [.065 sec, .171 sec] (Fig 6 top-right).

## Discussion

Experiment 1 found the sequential patterns in sequences of decisions as predicted by the attractor model, namely oscillatory switching and perseveration. It also found the predicted RT effects for choosing the preferred option and for staying with the preferred option. However, one could criticize in Experiment 1 that the sequences consisted of identical choice option in distance and values. While distances might have been occluded by the repositioning of options from trial to trial, the constancy of values could obviously be detected by participants. We hence performed experiment 2 to mitigate this argument.

## Experiment 2

Experiment 2 aimed to replicate the findings of experiment 1, but used longer sequences of choices and aimed to mask the similarity of these choices by superficially varying option's values. Despite these changes, we expected to find the predicted patterns of oscillatory switching (Hypothesis I) and perseveration (Hypothesis II) and the RT effects for choosing the preferred option (Hypothesis III) and for staying with the preferred option (Hypothesis IV).

### Methods

**Ethics statement.** The study was performed in accordance with the guidelines of the Declaration of Helsinki and of the German Psychological Society. An ethical approval was not required since the study did not involve any risk or discomfort for the participants. All participants were informed about the purpose and the procedure of the study and gave written informed consent prior to the experiment. All data were analysed anonymously.

**Participants.** 20 participants of the Technische Universität Dresden took part in the experiment. All participants had normal or corrected-to-normal vision. They gave informed consent to the study and received either payment or class credit.

**Procedure and setup.** The procedure and setup was similar to Experiment 1, with the following exceptions.

Experiment 2 was split in two runs of the game. In the first run, we measured the indifference points of each participant across choice option varied along the different distances and values (see Design). In the second run, we used these estimated indifference points to adjust our manipulation of the options' relative attractiveness in the sequences of choices. We presented sequences of 30 trials for each relative attractiveness configuration. Since the sequences were much longer than in Experiment 1, we made each single trial within a sequence superficially different by randomly varying the value of the LF option. Beneath this superficial difference in absolute value, however, we kept the subjective values of the SN and the LF option constant: We chose the SN value relative to the random LF value according to the estimated indifference point plus or minus an additional amount to influence the SN options' attractiveness. Relative to the indifference point, a sequence could then consist of SN options that were more attractive, equally attractive, or less attractive.

**Design.** In run 1, we measured the indifference points of each participant. We presented choices with 3 intervals ($D_N = 2$, 3, or 4 fields, chosen randomly; $D_F = D_N + 2$, 4, or 7 fields) and 8 differences in value ($V_L = [50 - 99]$ credits, chosen randomly; $V_S = 20$, 40, 60, 70, 80, 85, 90, or 95% of $V_L$).

In run 2, we used this estimated indifference point to adjust our manipulation of the options' relative attractiveness for sequences of 30 trials. Within a sequence, we kept the distances of the two options constant, chosen from 3 soon times ($D_N$ = 2, 3, or 4 fields, randomly chosen for each sequence) and 3 intervals ($D_F = D_N$ + 2, 4, or 7 fields, systematically chosen across sequences). Within each sequence, we randomly chose the LF option's value ($V_L$ = [50 − 99] credits). We kept the subjective values of the SN and the LF option constant relative to the indifference point by choosing the SN value relative to the random LF value. Each sequence could hence consist of SN options that were more attractive (indifference point -20%), equally attractive (indifference point +/-.001%) or less attractive (indifference point +20%). This resulted in 12 sequences of 30 trials each; 360 trials for a complete block.

## Results

Hypothesis I stated that in sequences of trials that start with a choice of the attractive option, we expect no switches or an even number of switches. In contrast, in sequences of trials that start with a choice of the unattractive option, we expect an odd number of switches. For trials with no clearly attractive option, we expected a random distribution of switches unbiased to an odd or even number of switches.

In this experiment, we used sequences of options in the experimental block that were already adapted to participants' indifference points as measured in the measurement block. To establish comparability to experiment 1 in the number of switches, we split up the sequences of 30 equal configurations of options into 5 subsequences of 6 consecutive options. For these subsequences, we counted the number of sequences that started either with the choice of the SN option or the LF option and that yielded either an even or an odd number of switches between options. Again, for statistical analysis, we subtracted the number of odd switches from the number of even switches. The resulting difference score was analysed in a RMANOVA. This revealed neither a significant main effects of the initially chosen option (SN vs. LF), $F(1,19)$ = 3.845, $p$ = .065, $\eta^2_p$ = .168, nor a significant main effect of attractiveness (SN vs. neutral vs. LF), $F(2,38)$ = 3.358 $p$ = .065, $\eta^2_p$ = .150, but–as predicted–a significant interaction, $F(2,38)$ = 48.794, $p < .001$, $\eta^2_p$ = .720. Hence, we found the predicted pattern of a largely higher number of even switches for sequences in which the attractive option was chosen initially.

Hypothesis II stated that we should find a perseveration bias, which is that the probability to choose the option again that has been chosen previously, is higher than the probability to switch between options. We again performed a Markov analysis across all trials, in which we counted the number of SN and LF choices with respect to the previously chosen option. For statistical analysis, we analysed whether the difference of switch and stay probabilities was positively different from zero, using a dependent samples $t$-test, which revealed a significant difference from zero, $t(19)$ = 10.953, $p < .001$, $d$ = 2.449. Hence, a clear perseverative bias was present as predicted by the model. A graphical summary of the results is given in Fig 5b.

Hypothesis III expected faster RT for trials in which participants chose the preferred option. We performed a t-test on the dependent variable RT with the independent variable *choice of preferred-unpreferred option* in trials with one attractive option. It yielded no significant difference, $t(19)$ = 1.01, $p$ = .328, 95% CI [-.033 sec, .095 sec], though the effect pointed in the expected direction (Fig 6 bottom-left).

Hypothesis IV stated that trials in which participants stayed with the preferred option should be faster than when participants switched to the preferred option after choosing the unpreffered option in the previous trial. We performed a t-test on the DV RT with the independent variable *stay with/switch to preferred option* in trials with one clearly attractive option.

It yielded a significant difference, $t(19) = 3.48$, $p = .0025$, 95% CI [.028 sec, .113 sec] (Fig 6 bottom-right).

## Discussion

Experiment 2 replicated Experiment 1 and the predictions by the attractor model for choice patterns and partly for RT, despite longer sequences and superficially different options. Though RT data did not yield the significant difference for choices of the preferred option, the descriptive difference pointed in the right direction.

   With these two successful validations, only the third of the model's predictions still remains to be validated:, a modulation of perseveration by the length of the delay between two consecutive decisions. To this end, we reanalysed data from a study that is already published ([21], https://osf.io/23ax5/).

## Reanalysis of existing data

The attractor model predicts that perseveration will be affected by the delay between two consecutive decisions. We reanalysed data from a previously published study that had used the same paradigm, though with a different mode of varying the presented option's values. The original study had used sequences of ascending or descending differences between the two offered options. Each sequence had been presented one time with a short delay (the inter trial interval between two trials) or a long delay.

   Though the setup using ascending or descending sequences is different to the repeated presentation of options in experiments 1 and 2, it still allows for an analysis of the consequences of different delays, when the analysis focusses on the mid-sequence trials for the level of perseveration that should occur. For these trials, the model's prediction leads to the hypothesis that perseveration (as indicated by markov analysis) is stronger in trials with a short delay compared to trials with a long delay between trials.

## Methods

   **Ethics statement.**   The study was performed in accordance with the guidelines of the Declaration of Helsinki and of the German Psychological Society. An ethical approval was not required since the study did not involve any risk or discomfort for the participants. All participants were informed about the purpose and the procedure of the study and gave written informed consent prior to the experiment. All data were analysed anonymously.

   **Participants.**   59 participants of the Technische Universität Dresden took part in the experiment. All participants had normal or corrected-to-normal vision. They gave informed consent to the study and received class credit for their participation.

   **Procedure and setup.**   The procedure and setup was similar to Experiment 1. However, participants worked on sequences of eight trials in which the distance of the LF option was increased or decreased by one field from trial to trial (for exact values, see design). Furthermore, the inter trial interval (ITI) between two trials was varied across sequences (1 second or 2.5 seconds, compared to 1.3 second in experiments 1 & 2). Participants were not informed about both, the sequences and the varying ITI. Finally, participants were randomly split into two groups, a positive mood and a negative mood group, which affected the background in the game and pictures that participants were shown between blocks.

   **Design and data selection.**   The options' values ranged from one to ten credits, with the values of the two options in each trial adding up to eleven credits to keep the overall value of each trial constant (smaller/ larger reward pairs were: 1/10, 2/9, 3/8, 4/7, 5/6). The SN option

was two or three fields away from the avatar and the LF was an additional 1, 2, 3, 4, 5, 6, 7, or 8 fields further away from the avatar than the SN option.

To stay with the type of analyses we used hitherto, we selected the inner four trials of each sequence, i.e. the 3rd, 4th, 5th, and 6th trial. Within these trials, one could expect participants to stay with or to switch away from the option that they had initially chosen at the start of the sequence of trials, while one could assume participants to stay with the chosen trial in the first 2 and the last 2 trials of a sequence. This logic holds for both, ascending and descending trials so that we can collapse across this variable. We further collapsed across the mood groups to increase the power of the applied markov analysis.

Within this setup, participants completed 601.81 trials on average (SD = 116.45 trials). Considering the selection of the inner 4 trials of each sequence (1/2 of the trials of each sequence) and the 2 ITIs, this results in an average of 150 trials per participant in the markov analysis.

## Results

Our hypothesis stated that we should find a perseveration bias, which is that the probability to choose the option again that has been chosen previously, is higher than the probability to switch between option. This perseveration bias should be larger for the short ITI compared to the long ITI. We again performed a Markov analysis for all included trials, in which we counted the number of SN and LF choices with respect to the previously chosen option. For statistical analysis, we analysed whether the difference of switch and stay probabilities was different for the short and the long ITI, using a dependent samples $t$-test, which revealed a significant difference, $t(58) = 3.041$, $p = .004$, $d = .396$. Hence, a difference in perseverative bias was present as predicted by the model.

## Discussion

The reanalysis shows that the delay between two consecutive choices influences the strength of perseveration as predicted by the model. Even though the reanalysis used data from the same paradigm, the setup of trials was different to the one in experiments 1 and 2 and hence not optimal to find the predicted difference. This might explain why the effects are relatively small, though still highly reliable in the studied sample.

## Simulation study on an intertemporal choice questionnaire

One might ask whether the incorporation of sequential patterns has substantive implications for real-life research. If we consider that sequential phenomena like perseveration influence each trial in a sequence value-based decisions, then ignoring these phenomena might lead to inaccurate conclusions. For example, the monetary choice questionnaire [28] has been used to study differences between control groups and groups with substance use disorder, i.e. heroin addicts [29, 30]. Higher $k$-values (the curvature parameter of the hyperbolic discounting model, see [31]) in such a study are usually interpreted as the heroin addict group showing more impulsiveness. However, since the questionnaire doesn't factor in perseveration effects, the conclusion of a difference between the two populations that lies in the discounting itself might be premature.

We simulated choices in the questionnaire with our model for a $k$-value of 0.01 (resulting in 44% of choices for the SS option according to the questionnaire). We performed two runs of the simulation with exactly the same model configuration. The only difference was that in the first run we used the short delay between decisions, as used in the other simulations here, whereas in the second run, we used a long delay to eradicate any perseveration effects. The latter run hence implements the assumption of independent single decisions.

When delays between decisions were long enough to eradicate any perseveration, the model choose exactly as the original *k*-value of 0.01 predicts. However, for the short delay, the *k*-value rises to a value of 0.015 due to the specific order of choices influencing each other. Hence, the fact that a heroin population shows higher *k*-values might not (or not only) be attributable to more impulsive decision making, but instead could be caused by hastier transitions from item to item leading to perseveration. These two interpretations, a change in the decision process and weighing of the options (as usually assumed) vs. a change in perseveration (as shown here), are largely different and lead to different conclusions about disorder-related changes in decision making that go beyond a purely academic question.

## General discussion

We proposed to extend the current trial-oriented focus of value-based decision research to capture sequential patterns of decision making across decisions [11–16]. For this purpose, we considered the inherent process dynamics that are typical for non-linear coupled models, such as leaky integrator models [23], parallel constraint satisfaction models [32], and–as a very simple instance used here–neural-inspired attractor models [11]. This model takes into account how choices influence each other and how such interactions give rise to patterns in sequences of decisions that reflect effects of memory traces of previous choices, which are modulated by noise in the system. Using computational simulation, we derived three typical patterns of decision making and two RT effects. We validated these predictions in two experimental studies that were specifically designed to study sequential patterns, and via reanalysing data from a third study. All studies used a specific decision game that allows investigating the dynamics of delay discounting in an immersive setting and occluding sequences of similar choices.

Though we showed that the predicted patterns can be found in this specific setup, the model's assumptions about inherent dynamics are generic. Not only have such effects been found before in the domain of perceptual decision-making [15, 23, 33–35]. Especially perseveration and even higher order sequence effects have been extensively studied in the context of perceptual decision making [e.g. 17–19] and have been modelled based on attractor dynamics [36]. However, such effects are often neglected in studies of value-based decision making, where trials of interest are analyzed in isolation or–as we show as even worse–the sequence of trials is constant in the case of questionnaires, e.g. the monetary choice questionnaire. We show that the common assumption from perceptual decision making should also be applied in value-based decision making, which is, that the visual dominance of different percepts (in perceptual decision making) or the different attractiveness of the options (in value-based decision making) governs the decision process, but the information about this dominance or attractiveness is noisy and sticky which can lead to switches or perseveration in sequences of decisions and thereby may lead to choices of a weaker or suboptimal option [11]. Hence, our work could be seen as a generalization of the work from perceptual decision making to value-based decision making and perseveration is one example of first-order sequence effects (for a more extensive overview of sequence effects and the assumed underlying mechanisms in perceptual decision making, see [17]).

Concerning the research process in value-based decision making, our work also delivers an interesting side-note: The decisive difference between dynamic models (LCA, PCS and the attractor model presented here) and common sequential sampling models (LBA and diffusion model) is their inherent dynamics [35], which allows for the sequential patterns to emerge by themselves. Sequential sampling models could still mimic such effects by explicitly adding a bias that is influenced by the previous decision, e.g. by adding a starting bias of 10% of the previously chosen option's activation. However, this would be a newly added process that is

different to the already existing constant starting bias in the direction of a specific option (independent of the previous choice). Hence, sequential sampling models in their standard form cannot *predict* such data by themselves since it is not specified what happens in these models between decisions and only adding a specific mechanism could make them able to this. In contrast, the activation dynamics intrinsic to dynamic models offer this mechanism by themselves. The creators of common sequential sampling models removed these–usually non-linear–dynamics on purpose. For example, the ballistic linear accumulator model [37] was conceptualized as a simplification of the leaky accumulator model [23]. The purpose of the simplification was to allow for easier quantitative fitting of response time data. Our findings indicate that, while such simplifications might be necessary for technical reasons, one should not lose sight of phenomena for which simplified models lose predictive power [35].

## Limitations

We want to discuss three limitations about the predictions/modelling work of this study.

First, it is an open question, whether our model draws an adequate picture of the decision making process for the delay discounting decisions studied here, since these decisions are multi-attributive while our model only receives one input per option. Though it is an open question whether the decision system indeed uses a single variable, i.e., the subjective value used in our model or bases its decision indeed on the single attributes, two points indicate that this obvious limitation does not question the general point we make here. First, our model represents a class of models with inherent dynamics which includes more complex, multi-attributive models, e.g. the PCS model [32]. For this model, one can expect the same effects that we showed here. Second, and more practically, an increasing number of works using sequential sampling models to model delay discounting (e.g. [8, 9]) are based in the same assumption of one input per option. Our work proceeds exactly on this path and extends the work based in sequential sampling models, so that the number of inputs to the model is not central to the main point of our study.

Second, the effects predicted here, are in their nature parameter dependent, but we refrained from presenting a complete parameter exploration, for two reasons. First, it would be trivial since the ITI dependence predicted by the model itself indicates that a stronger decay or an even longer ITI would cancel the sequential effects shown here. Second, we used parameters for our model that we kept constant across several publications and hence we see the model as a complete unit for predicting different qualitative patterns of value-based decision making across several studies. We hence think that it speaks to the strength of the model that we do not have to adapt parameters from study to study but instead have a simple model that generalizes across studies.

Third and related to the second point, one could speculate whether the predicted effects occur only when using our specific model with its specific activation formulas. However, as the literature on perceptual decision making also indicates, sequential patterns and the specific contribution of the activation dynamics are common across all models that implement such dynamics, be it leaky integrator models [23] or parallel constraint satisfaction models [24, 25] or other models [17, 19]. Since these effects are ubiquitous, it is all the more astonishing that empirical research in value-based decision making usually ignores the potential caveats associated with them.

## Conclusion

In this work, we show how integrating sequential patterns of values-based decision-making into the research program may substantially further our understanding of the behavioural and

neural processes that underlie optimal and suboptimal decision-making. Considering the stability of general patterns in a sequential value-based decision-making perspective may also improve upon the explanatory power of single decisions that suffer from a high level of unexplained noise instead. We speculate that the incorporation of a sequential perspective into decision models and empirical studies will support an understanding of deficient decision-making and the underlying circuitry in the healthy and the dysfunctional brain with respect to the stability of decision patterns.

## Supporting information

**S1 Appendix. Computational modelling and simulation.**
(DOCX)

## Acknowledgments

We thank Thilo Gross for his comments on previous versions of the manuscript and on the design and data-analysis during the study.

## Author Contributions

**Conceptualization:** Stefan Scherbaum, Steven J. Lade, Stefan Siegmund, Maja Dshemuchadse.

**Data curation:** Stefan Scherbaum, Steven J. Lade, Maja Dshemuchadse.

**Formal analysis:** Stefan Scherbaum, Steven J. Lade, Maja Dshemuchadse.

**Investigation:** Stefan Scherbaum.

**Methodology:** Stefan Scherbaum, Steven J. Lade, Maja Dshemuchadse.

**Project administration:** Stefan Scherbaum, Maja Dshemuchadse.

**Software:** Stefan Scherbaum.

**Validation:** Maja Dshemuchadse.

**Writing – original draft:** Stefan Scherbaum, Steven J. Lade, Stefan Siegmund, Thomas Goschke, Maja Dshemuchadse.

**Writing – review & editing:** Stefan Siegmund, Thomas Goschke, Maja Dshemuchadse.

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
