## [Decision Letter · Decision Letter 0]

2 Dec 2021

PONE-D-21-29270From single decisions to long-term choice patterns: Extending the dynamics of value-based decision-makingPLOS ONE

Dear Dr. Scherbaum,

Thank you for submitting your manuscript to PLOS ONE. After careful consideration, we feel that it has merit but does not fully meet PLOS ONE’s publication criteria as it currently stands. Therefore, we invite you to submit a revised version of the manuscript that addresses the points raised during the review process. Both reviewers ask to be clearer about the presentation of the model, what it predicts, and how it compares to other models. Even though novelty is not per se a publishing criterion for the journal, it is relevant to state how your model compares to other models (e.g., the sequential sampling models mentioned by reviewer 2) in accounting for the current data.

We look forward to receiving your revised manuscript.

Kind regards,

Tom Verguts

Academic Editor

PLOS ONE

Journal Requirements:

 “SSCH: German Research Council (DFG) (grant SFB 940/3 2020 Project A8)

SL: Swedish Research Council Formas (Project Grant 2014-589).”

Additional Editor Comments (if provided):

Small comments:

- line 131: “compared to stronger perseveration” —> remove?

- p 7: I think the terms SN and LF are not yet explained here.

- Indifference point; can this change due to learning?

- Odd and even switches; can’t you just analyse the number of odd switches? The number of even switches seems redundant (even though it may not fully be so here, due to the variable number of trials. Similarly, in Figure 5, the number of stays and switches seem redundant to me (just one variable can be analyzed).

- p 14, line 289: you mean than in Experiment 1?

Reviewers' comments:

Reviewer's Responses to Questions

**Comments to the Author**

1. Is the manuscript technically sound, and do the data support the conclusions?

Reviewer #1: Partly

Reviewer #2: Partly

2. Has the statistical analysis been performed appropriately and rigorously? 

Reviewer #1: Yes

Reviewer #2: Yes

3. Have the authors made all data underlying the findings in their manuscript fully available?

Reviewer #1: Yes

Reviewer #2: Yes

4. Is the manuscript presented in an intelligible fashion and written in standard English?

Reviewer #1: Yes

Reviewer #2: No

5. Review Comments to the Author

Reviewer #1: Scherbaum et al. propose a simple LCA model and investigate how the temporal dynamics of the model account for “long-term” effects of decision history on current decisions. In particular, their model explains: 1. switch patterns (as a function of attractiveness) during a sequence of decisions, 2. perseverance, i.e. increased probability to select the same choice if selected previously, and 3. temporal dependency, i.e. increase probability of switching/staying if ITI is long/short. Furthermore, they predict RT effects, such that choices with a strong preferred options should be made faster and that repeated trials also should be made faster versus switch trials.

While I am fairly sympathetic to this study and I believe the analyses are well explained and nicely conducted, several issues need to be addressed before I can recommend publication.

Major comments:

**My main concern is with the lack of clarity when describing the theoretical novelty of this work. The authors should clearly delineate what their model predicts above and beyond previous models of trial history effects, being at the perceptual or value-based domain. The LCA was developed to account for perceptual choices. So if the novelty of this work resides in analyzing the effects of residual activation in the system on the next trial, it should be clarified and systematically compared to previous models (e.g., the Gao model). In sum, the authors should clearly state what is novel (or not) in their work and how the collected data sets allow to arbitrate between theirs and existing models.

**Linked to the previous comment, it seems that all the proposed effects can be explained as a function of residual activity in the network at the moment of the next trial. This implies that specific parameters are crucial to account for the data, i.e. the leak, the noise, their tradeoff… If all the effects can be explained by the leak (which it seems), then this should be clarified. Exploration of the parameter space is completely lacking in the current version of the paper. Furthermore, the authors would want to investigate whether their conclusions hold with more biologically plausible activation functions like the ReLU, i.e. non-linearity at zero activation. As it is, the fact that the non-preferred unit dips below zero may convey a non-biological advantage to the model in order to account for the data.

**The authors should clarify in the introduction what they mean by “long-term” effects. The dynamics of LCA models and their impact on choice sequences may be more defined as short-term effect rather than long term effects. While I see how associative attractor networks may display long-term effects, it is unclear how long is the “long-term” effects of LCA models.

**Intuitively, it seems that if the choice sequences get longer, i.e. 100 trials, you would observe an increasing drift in the stay probability. Did the authors investigate the model’s behavior with longer sequences and maybe reveal emergent properties of their model in other settings (that can be tested later on)?

**It is not clear how the predictions in Figure 3 are not already captured by those in Figure 2B? At the very least they could be combined. If the probability of stay is higher, it is because residual activation is already high for that choice, and hence the choice will be made faster (under the assumption of a fixed threshold). Also, I fail to see the novelty in these RT effects, couldn’t these effects be captured by the drift and the bias parameter, respectively – as shown in many previous studies?

Minor comments:

Line 68: “Since such decisions are about the weighing of an options value and other properties, such decisions are called value-based decisions.” It reads as if you compare the value of an option to other properties. Whereas as these properties (time, rewards, effort…) are usually integrated into a cost-benefit evaluation whose outcome is the value for that offer. Maybe the authors can rephrase or better explain what they mean here.

Line 77: “found to be ubiquitous in perceptual decision making [e.g. 17–19]”. Another study which also focuses on the effect of previous trial on current decisions deserves to be cited here, Aben, Verguts, Van den Bussche JEP:HPP (2017).

Line 81: “…which is why these models can be used to make predictions which long-term patterns one could expect in value-based decision tasks”. This sentence is confusing. Do the authors mean that LCA and associative attractor networks can be used to investigate long-term, contextual, effects in value-based decision making? As it is, the sentence is not clear and should be rephrased.

Line 129: “The third pattern is a temporal dependency of perseveration: Separating two decision by a short delay compared to a longer delay should lead to weaker perseveration compared to stronger perserveration (see Figure 2b).” I believe the authors meant to inverse the contingencies, i.e. short delay leads to stronger perseveration.

Typos:

In general, I recommend the authors to have a careful look at their paper for typos and grammatical issues.

Line 37: value-based

Line 68: weighing of an option

Line 129: The third pattern is a temporal dependency of perseveration: Separating two decision

Reviewer #2: In this paper the authors report a set of “long-term” patterns in value-based choices and explain those using an attractor choice model. There are interesting aspects in these study, however several methodological details need to be better explained. Additionally, it is not clear whether the attractor model differs substantially relative to sequentially sampling models, that are typically used in order to explain choice history biases.

1) As the authors mention, the odd/ even number of switches prediction could also arise from sequential sampling models or any choice model in sequences where there is a preferred alternative. It is not clear though if this hypothesis/ prediction is worth discussing.

2) It is not clear whether the perseverance prediction could be made by a sequential sampling model. At the moment the details about the perseveration analysis are scarce and it is not straightforward to understand how the LF/SN ratio was taken into account, and whether the perseverance bias occurs naturally merely due the fact that in most blocks there was a preferred option. Is the bias present also in neutral sequences?

3) It is not mentioned how RT’s in this task are calculated. Are RT’s normalised for the distance travelled on each trial?

4) Details about the measurement block, in which the indifference points are calculated, are missing.

5) More generally, it would be useful to contrast the attractor model with a sequential sampling model in which choice history biases arise either due to changes in the starting point or the drift rate of evidence accumulation. Otherwise, it is not clear whether the theoretical contribution of this paper is novel.

6) It is not clear whether the biases reported here are “long-term” or simply arising between the current and the previous (n-1) trial.

7)About the questionnaire simulation, is the higher kappa found under short ITI specific to a certain order of questions? Could, for instance, a different order result in lower kappa under short ITI.

8)The manuscript has several typos and could benefit for more thorough proof reading.

6. PLOS authors have the option to publish the peer review history of their article (what does this mean?). If published, this will include your full peer review and any attached files.

Reviewer #1: No

Reviewer #2: No

---

## [Author Response · Author response to Decision Letter 0]

24 Mar 2022

Dear Tom Verguts,

we thank you and the reviewers for the very helpful comments on our manuscript. 

In our revised version, we incorporated all of yours and the reviewers’ suggestions and are happy that in our view this greatly improved and completed the manuscript.

In this letter, I will go through the various points in the order in which they were raised in the reviews. Our responses are printed in bold and start with a dash. Changes in the manuscript are marked in red font.

General remarks

1. When reading the reviewers’ comments, we had to realize that our original writing caused a substantial misunderstanding about the focus of the paper. We are very grateful to the reviewers for clearly pointing out that they thought the focus was on a new model. Instead, we aimed to show that sequential choice patterns are present in value-based decision making as they are well known in perceptual decision making. Our main point was that expectations of such patterns arise naturally when one considers the activation dynamics of options between trials and that it is necessary to consider such patterns also in studies of value-based decision making, especially when using questionnaires with fixed decision sequences to identify individual differences. We rewrote large parts of the manuscript to rectify this misunderstanding, strengthening the focus on the data and more clearly stating that our model is simply a simple instance of the models used previously especially in perceptual decision making.

2. We were fully aware that such sequential effects (and more long-term higher order sequential effects) had been show for perceptual decision making (and we cited several of them in the introduction). In fact, it was our aim to illustrate and stress that value-based decision making should follow and consider these original findings since such patterns are also present in value-based decision making. We hence rewrote parts of the manuscript to refer more clearly to the findings from perceptual decision making that our study builds on.

Journal Requirements:

Done

 “SSCH: German Research Council (DFG) (grant SFB 940/3 2020 Project A8)

SL: Swedish Research Council Formas (Project Grant 2014-589).”

This is correct: The funders had no role in study design, data collection and analysis, decision to publish, or preparation of the manuscript.

The repository will be made publicly accessible on the note of acceptance.

Done

Additional Editor Comments (if provided):

Small comments:

- line 131: “compared to stronger perseveration” —> remove?

Done, removed.

- p 7: I think the terms SN and LF are not yet explained here.

We removed this.

- Indifference point; can this change due to learning?

Yes, it can in principle. The indifference point should – in our view – be seen as a measure derived from choice behavior. If discounting changes, then, of course, the indifference point also changes.

- Odd and even switches; can’t you just analyse the number of odd switches? The number of even switches seems redundant (even though it may not fully be so here, due to the variable number of trials. Similarly, in Figure 5, the number of stays and switches seem redundant to me (just one variable can be analyzed).

This is true in principle. However, when presenting the data, we gained the impression that most people found it easier in the somewhat redundant version used here.

- p 14, line 289: you mean than in Experiment 1?

Yes, sorry. It’s corrected.

5. Review Comments to the Author

Reviewer #1: 

Scherbaum et al. propose a simple LCA model and investigate how the temporal dynamics of the model account for “long-term” effects of decision history on current decisions. In particular, their model explains: 1. switch patterns (as a function of attractiveness) during a sequence of decisions, 2. perseverance, i.e. increased probability to select the same choice if selected previously, and 3. temporal dependency, i.e. increase probability of switching/staying if ITI is long/short. Furthermore, they predict RT effects, such that choices with a strong preferred options should be made faster and that repeated trials also should be made faster versus switch trials.

While I am fairly sympathetic to this study and I believe the analyses are well explained and nicely conducted, several issues need to be addressed before I can recommend publication.

Major comments:

**My main concern is with the lack of clarity when describing the theoretical novelty of this work. The authors should clearly delineate what their model predicts above and beyond previous models of trial history effects, being at the perceptual or value-based domain. The LCA was developed to account for perceptual choices. So if the novelty of this work resides in analyzing the effects of residual activation in the system on the next trial, it should be clarified and systematically compared to previous models (e.g., the Gao model). In sum, the authors should clearly state what is novel (or not) in their work and how the collected data sets allow to arbitrate between theirs and existing models.

We thank the reviewer for this clear comment as it indicates a misunderstanding that we caused by our presentation of our study. As we indicate in the general comments, we did not mean to implicate that we use a new or especially innovative model here. While we already cited LCA and PCS work, we understand that our original writing might have implicated this. We hence rewrote the respective sessions in the introduction (p. 4 ) and discussion (p. 23) to make clear that our model is simply a very simply instance of the aforementioned models that implements activation dynamics between trials. The new value of our work is to transfer the well-established findings from perceptual decision making to value-based decision making and to show – primarily – that patterns that are expectable are indeed present in value-based decision making and can even lead studies in this field astray if not considered.

**Linked to the previous comment, it seems that all the proposed effects can be explained as a function of residual activity in the network at the moment of the next trial. This implies that specific parameters are crucial to account for the data, i.e. the leak, the noise, their tradeoff… If all the effects can be explained by the leak (which it seems), then this should be clarified. 

This is simply true and we are sorry if our writing might have implied more. All these patterns are expectable when you consider between trial activation dynamics (the leakage). However, in value-based decision making these dynamics are usually not considered and this is what we want to change with the presentation of our findings. We adapted our terminology to make this focus clearer, from long-term patterns to sequential patterns, throughout the manuscript and in the title and we hope that the changed sections in the intro and the discussion make this now clearer.

Exploration of the parameter space is completely lacking in the current version of the paper. 

This is true and there is a reason for it: We use the same parameterization as we used across several papers. It is clear that the effect depends on the decay (as we show in the prediction of ITI dependence. Hence, exploring the parameter space is in our view expendable in this case. We included this point in the limitations section (limitation 2, p. 25).

Furthermore, the authors would want to investigate whether their conclusions hold with more biologically plausible activation functions like the ReLU, i.e. non-linearity at zero activation. As it is, the fact that the non-preferred unit dips below zero may convey a non-biological advantage to the model in order to account for the data.

As the reviewer states himself, the decay effect is quite common for all models that implement activation dynamics. For example, the PCS model uses quite different formulas but still comes to the same conclusions (and even a diffusion model if one builds a decay of the previous activation in the model, see the simulations to answer reviewer 2). Since our main aim is not to present a new decision model, but to show that even a very simply model of activation dynamics can predict sequential patterns in value-based decision making and that these patterns are indeed present in empirical data, we refrained from adding any further model analyses. We mention this point now in the limitations section (p. 25).

**The authors should clarify in the introduction what they mean by “long-term” effects. The dynamics of LCA models and their impact on choice sequences may be more defined as short-term effect rather than long term effects. While I see how associative attractor networks may display long-term effects, it is unclear how long is the “long-term” effects of LCA models.

We are sorry for the confusion caused by our writing. We changed the term long-term patterns to sequential patterns and stress throughout the manuscript that our point is that these sequential patterns also present in value-based decision making even though superficial stimulus properties may vary in this domain.

**Intuitively, it seems that if the choice sequences get longer, i.e. 100 trials, you would observe an increasing drift in the stay probability. Did the authors investigate the model’s behavior with longer sequences and maybe reveal emergent properties of their model in other settings (that can be tested later on)?

Since the model also works with a decision threshold and then uses the decayed residual activation leading to the bias, the model cannot show additional biases across longer sequences. We are sorry if our usage of the term long-term patterns might have implicated this, which is why we changed even the title to “sequential patterns”.

**It is not clear how the predictions in Figure 3 are not already captured by those in Figure 2B? At the very least they could be combined. If the probability of stay is higher, it is because residual activation is already high for that choice, and hence the choice will be made faster (under the assumption of a fixed threshold). Also, I fail to see the novelty in these RT effects, couldn’t these effects be captured by the drift and the bias parameter, respectively – as shown in many previous studies?

We would agree with the reviewer, but in a previous reviewing process another reviewer wanted us to exactly show these predictions and check them in the data to gather convincing evidence that the predictions and patterns in the data are based in the same underlying mechanism. Since the patterns in choices are complex in their own right, we think that separating the RT patterns in an extra figure eases things for the reader – especially as number of figures shouldn’t be an issues in an online publication.

Minor comments:

Line 68: “Since such decisions are about the weighing of an options value and other properties, such decisions are called value-based decisions.” It reads as if you compare the value of an option to other properties. Whereas as these properties (time, rewards, effort…) are usually integrated into a cost-benefit evaluation whose outcome is the value for that offer. Maybe the authors can rephrase or better explain what they mean here.

Thank you. We changed this to “Since such decisions are about the weighing of the different attributes of options, e.g., their monetary value, their temporal delay or probability, and these attributes are assumed to be integrated into a common value for each option, such decisions are called value-based decisions.”

Line 77: “found to be ubiquitous in perceptual decision making [e.g. 17–19]”. Another study which also focuses on the effect of previous trial on current decisions deserves to be cited here, Aben, Verguts, Van den Bussche JEP:HPP (2017).

Thank you, we added the citation.

Line 81: “…which is why these models can be used to make predictions which long-term patterns one could expect in value-based decision tasks”. This sentence is confusing. Do the authors mean that LCA and associative attractor networks can be used to investigate long-term, contextual, effects in value-based decision making? As it is, the sentence is not clear and should be rephrased.

We hope that our change to the term “sequential patterns” solved this.

Line 129: “The third pattern is a temporal dependency of perseveration: Separating two decision by a short delay compared to a longer delay should lead to weaker perseveration compared to stronger perserveration (see Figure 2b).” I believe the authors meant to inverse the contingencies, i.e. short delay leads to stronger perseveration.

Thank you, we corrected this.

Typos:

In general, I recommend the authors to have a careful look at their paper for typos and grammatical issues.

Line 37: value-based

Line 68: weighing of an option

Line 129: The third pattern is a temporal dependency of perseveration: Separating two decision

Thank you. We rechecked the manuscript and hope we could largely improve on this point.

Reviewer #2: In this paper the authors report a set of “long-term” patterns in value-based choices and explain those using an attractor choice model. There are interesting aspects in these study, however several methodological details need to be better explained. Additionally, it is not clear whether the attractor model differs substantially relative to sequentially sampling models, that are typically used in order to explain choice history biases.

1) As the authors mention, the odd/ even number of switches prediction could also arise from sequential sampling models or any choice model in sequences where there is a preferred alternative. It is not clear though if this hypothesis/ prediction is worth discussing.

We are sorry, but we already mention in the predictions section that this is a sanity check that the model acts as one would expect when combining option attractiveness and noise in a way that neither of both completely dominates the choice. 

2) It is not clear whether the perseverance prediction could be made by a sequential sampling model. At the moment the details about the perseveration analysis are scarce and it is not straightforward to understand how the LF/SN ratio was taken into account, and whether the perseverance bias occurs naturally merely due the fact that in most blocks there was a preferred option. Is the bias present also in neutral sequences?

First: a sequential sampling model could produce the effect if one integrates a systematic bias towards the last chosen option, e.g. by using 10% of the previous chosen option’s activation as starting bias for the following choice (see simulations below at second point). But – and this is the most important point – a SSM does not have the intrinsic activation dynamics that makes it PREDICT such an effect by itself, compared to all the other models (LCA, PCS, neural attractor model) that do have such dynamics and allow to trace the model behavior between trials. To make this work in a diffusion model, the bias needs to be set explicitely to a percentage of the previously chosen option’s activation, since a constant bias as implemented as a standard parameter in the diffusion model, would lead to a repetition bias that is independent of the previous choice. This is especially true for neutral trials as we will argument in the following. We stress this point now even more in the discussion (p. 23). 

Second: The neutral trials are, as the reviewer points out, the most critical test for the bias, even though the markov analysis is largely insensitive to the base rate of an option. So to be save, we looked only at these neutral trials in the real data, in the attractor model data, in data from a diffusion model without an explicitly modelled bias and in data from a diffusion model with an explicitly modelled biased (10% previous activation).

As can be seen in the figures below, the pattern is exactly as one would expect: The bias is visible in the data as predicted by the attractor model, it is present in the diffusion model with an explicitly modelled bias and not present in the diffusion model without bias.

Attractor model Data (Exp. 2)

Diffusion model with bias Diffusion model without bias

3) It is not mentioned how RT’s in this task are calculated. Are RT’s normalised for the distance travelled on each trial?

Thank you for this important point. We added this definition to the first occurrence of RT in the first results section. As in our previous work with this paradigm, we defined RT as the time from option presentation to the first click into a movement field. Hence, a normalization is not necessary. Notably, subjects only change their mind after this first click in a negligible proportion of trials, which are discarded for RT analysis if present (see Scherbaum et al., 2016, JDM).

4) Details about the measurement block, in which the indifference points are calculated, are missing.

As described in the methods, the measurement block worked exactly as described in experiment 1 using the options as described in the design section. We are sorry, but we do not see what can be added here.

5) More generally, it would be useful to contrast the attractor model with a sequential sampling model in which choice history biases arise either due to changes in the starting point or the drift rate of evidence accumulation. Otherwise, it is not clear whether the theoretical contribution of this paper is novel.

As mentioned above, we are sorry if our original writing suggested that we present a new model. Our point is that sequential patterns should be considered when studying value-based decision making, since they exist as shown previously for perceptual decision making. We use the attractor model as a simple instance of models with intrinsic activation dynamics that can PREDICT by itself such biases. A sequential sampling model can also produce such effects (as shown above), but you need to implement them explicitely because the model itself is only able to simulate the dynamics within a trial, i.e., the bias needs to be set to a percentage of the previously chosen option’s activation, since a constant bias would lead to a repetition bias that is independent of the previous choice. Although this might seem like a purely academic distinction, the fact that sequential patterns are almost ignored in value-based decision making research indicates that this is a real difference and our last simulation study even shows that ignoring such sequential patterns might lead studies of individual differences in value-based decision making astray. We hope that our rewriting of parts of the manuscript makes all this clearer now.

6) It is not clear whether the biases reported here are “long-term” or simply arising between the current and the previous (n-1) trial.

This is an important point and we changed long-term to sequential in the text.

7)About the questionnaire simulation, is the higher kappa found under short ITI specific to a certain order of questions? Could, for instance, a different order result in lower kappa under short ITI.

Yes, this is exactly the point. The questionnaires are used very often in studies of individual differences in value-based decision making and the static specific order of decisions is simply ignored by the researchers. This is why we simulated the results for this specific, very often used questionnaire with the standard order of items.

8)The manuscript has several typos and could benefit for more thorough proof reading.

We rechecked the manuscript and hope that we could improve the writing sufficiently.

---

## [Editor Report · Decision Letter 1]

6 Apr 2022

From single decisions to sequential choice patterns: Extending the dynamics of value-based decision-making

PONE-D-21-29270R1

Dear Dr. Scherbaum,

I think you have addressed the comments of the reviewers very well.

We’re pleased to inform you that your manuscript has been judged scientifically suitable for publication and will be formally accepted for publication once it meets all outstanding technical requirements.

Kind regards,

Tom Verguts

Academic Editor

PLOS ONE
---

## [Editor Report · Acceptance letter]

12 Apr 2022

PONE-D-21-29270R1 

From single decisions to sequential choice patterns: Extending the dynamics of value-based decision-making 

Dear Dr. Scherbaum:

I'm pleased to inform you that your manuscript has been deemed suitable for publication in PLOS ONE. Congratulations! Your manuscript is now with our production department. 

Kind regards, 

on behalf of

Dr. Tom Verguts 

Academic Editor

PLOS ONE